# Fungi Can Be More Effective than Bacteria for the Bioremediation of Marine Sediments Highly Contaminated with Heavy Metals

**DOI:** 10.3390/microorganisms10050993

**Published:** 2022-05-09

**Authors:** Filippo Dell’Anno, Eugenio Rastelli, Emanuela Buschi, Giulio Barone, Francesca Beolchini, Antonio Dell’Anno

**Affiliations:** 1Department of Marine Biotechnology, Stazione Zoologica “Anton Dohrn”, Villa Comunale, 80121 Naples, Italy; 2Department of Marine Biotechnology, Stazione Zoologica “Anton Dohrn”, Fano Marine Centre, Viale Adriatico 1-N, 61032 Fano, Italy; emanuela.buschi@szn.it; 3Institute for Marine Biological Resources and Biotechnology, National Research Council, Largo Fiera della Pesca 2, 60125 Ancona, Italy; giulio.barone@irbim.cnr.it; 4Department of Life and Environmental Sciences, Università Politecnica delle Marche, Via Brecce Bianche, 60131 Ancona, Italy; f.beolchini@univpm.it

**Keywords:** bioremediation, heavy metals, fungi, sediments, contamination

## Abstract

The contamination of coastal marine sediments with heavy metals (HMs) is a widespread phenomenon that requires effective remediation actions. Bioremediation based on the use of bacteria is an economically and environmentally sustainable effective strategy for reducing HM contamination and/or toxicity in marine sediments. However, information on the efficiency of marine-derived fungi for HM decontamination of marine sediments is still largely lacking, despite evidence of the performance of terrestrial fungal strains on other contaminated matrixes (e.g., soils, freshwater sediments, industrial wastes). Here, we carried out for the first time an array of parallel laboratory experiments by using different combinations of chemical and microbial amendments (including acidophilic autotrophic and heterotrophic bacteria, as well as filamentous marine fungi) for the bioremediation of highly HM-contaminated sediments of the Portman Bay (NW Mediterranean Sea), an area largely affected by long-term historical discharges of mine tailings. Our results indicate that the bioleaching performance of metals from the sediment is based on the addition of fungi (*Aspergillus niger* and *Trichoderma* sp.), either alone or in combination with autotrophic bacteria, was higher when compared to other treatments. In particular, fungal addition allowed obtaining bioleaching yields for As eight times higher than those by chemical treatments and double compared with the addition of bacteria alone. Moreover, in our study, the fungal addition was the only treatment allowing effective bioleaching of otherwise not mobile fractions of Zn and Cd, thus overtaking bacterial treatments. We found that the lower the sediment pH reached by the experimental conditions, as in the case of fungal addition, the higher the solubilization yield of metals, suggesting that the specific metabolic features of *A. niger* and *Trichoderma* sp. enable lowering sediment pH and enhance HM bioleaching. Overall, our findings indicate that fungi can be more effective than acidophilic autotrophic and heterotrophic bacteria in HM bioleaching, and as such, their use can represent a promising and efficient strategy for the bioremediation of marine sediments highly contaminated with heavy metals.

## 1. Introduction

Heavy metal (HM) contamination of marine sediments is a widespread environmental problem, particularly frequent in coastal areas subject to high anthropogenic impact (e.g., industrial practices, ore mining, dumping of elevated metal waste, excessive use of chemical fertilizers, sewage discharge) and reduced hydrodynamic regimes [1,2]. High concentrations of HM accumulate in sediments and can be transferred through the food web up to the higher trophic levels, with potential negative consequences on ecosystems and human health [3,4,5]. Indeed, HMs can determine oxidative stress and interference with protein folding and physiological functioning in vertebrates and invertebrates, causing several cellular/tissue disorders [6,7].

In the last decade, different approaches have been developed to reduce HM concentrations and their potential toxicity in the sedimentary matrix, with the purpose of alleviating marine sediment management costs and environmental impacts [8,9]. Indeed, the identification of effective treatments for the decontamination of HM-contaminated marine sediments can allow the transformation of sediments into a valuable resource to be used for different applications (e.g., building materials, beach nourishment, agriculture applications; [10,11,12,13]).

The main ex-situ approaches for the treatment of contaminated sediments (e.g., after dredging) include chemical-physical, thermal, and biological treatments [14,15,16,17,18,19,20,21]. The chemical-physical treatments can be used to (i) promote the solubilization of metals from the sediments by aqueous solutions containing chemical or chelating agents and by electrochemical processes; (ii) reduce the mobility of contaminants through complexation with stabilizing agents (like lime or cement, [8,22]). Thermal treatments can be used either to desorb HM from the sediment or to induce their immobilization into the sedimentary matrix [19,23]. Nevertheless, all these approaches are limited mainly due to high economic costs, low specificity, and the generation of large amounts of toxic wastes [24,25,26,27].

In the last years, biological strategies have received great attention for their major environmental compatibility and lower costs [5,28,29,30,31]. Two main strategies exist for the bioremediation of HM-contaminated sediments, which involve the use of microorganisms either to immobilize/stabilize or to mobilize/extract HMs [32,33]. While the first approach can help reduce HM toxicity by decreasing HM mobility (but leaving HMs within the sediments), the second approach can be preferred if the final aim is to remove HMs, hence effectively decreasing their concentrations in the sediments [20]. This latter process for HM decontamination can be achieved through the addition of microbes (especially bacteria and/or fungi) into the sediments (i.e., bioaugmentation) to promote HM bioleaching/solubilization and thus their mobilization and final removal from the sediments [10,34,35].

Chemoautotrophic Fe/S oxidizing bacteria like *Acidithiobacillus ferrooxidans*, *Acidithiobacillus thiooxidans*, and *Leptospirillum ferrooxidans* are well known to promote HM solubilization in marine sediments by oxidizing sulphur and/or Fe under acidic conditions and using O_2_ as a terminal electron acceptor [36,37,38,39,40,41,42], with the addition of elemental sulfur triggering higher HM removal efficiency [43,44,45,46]. The ability to solubilize metals from contaminated marine sediments is also known for acidophilic heterotrophic bacteria belonging to the genus *Acidiphilium* (e.g., *A. cryptum*) [39,41,42,47,48], which co-respire oxygen and ferric iron (Fe^3+^) by reducing Fe at low pH to oxidize organic substrates [49,50,51]. Some studies have also investigated the metabolic interactions between aerobic acidophilic Fe-reducers and Fe/S autotrophic oxidizing bacteria, highlighting significant synergies that can further enhance HM bioleaching efficiency, also in marine sediments [10,52,53,54,55,56].

Besides bacteria, also fungi are known to leach metals (thus enhancing HM removal from the contaminated matrix) through the production of organic acids (e.g., citric, oxalic, fumaric, and gluconic acids), which increase HMs solubility by lowering pH and forming water-soluble complexes with HMs [30,34,57,58,59,60,61]. However, to our knowledge, no studies exist to date on the use of marine-derived fungi for the removal of HMs from marine sediments through bioleaching, despite promising evidence on other matrixes such as contaminated soils or solid wastes [59,62] and freshwater sediments [63,64,65]. Moreover, only a few of these studies have contextually compared bacterial versus fungal HM bioremediation efficiency or tested the use of mixed consortia of bacteria and fungi [33,63], and never on marine sediments [41,42].

In the present study, we conducted laboratory experiments to contextually compare for the first time different treatments for the remediation of marine sediments heavily polluted by HMs, selecting Portman Bay as a pilot study area [66]. In detail, we investigated the solubilization efficiencies of some of the major HMs found in the sediments (Zn, Cd, and As) by comparing either the additions of chemicals alone or different microbial-based amendments (i.e., additions of chemo-autotrophic Fe/S oxidizing bacteria, chemo-heterotrophic bacteria, and fungi, either alone or in combination).

## 2. Materials and Methods

### 2.1. Study Area and Sampling

Sediment samples were collected using multiple corers in the Portman Bay (NW Mediterranean Sea) at a depth of about 43 m (37°34.070′ N, 0°50.659′ W). This area has been subjected for several decades (1957–1990) to the discharge of large quantities of waste materials (estimated at about 57 million tons) deriving from mining activities, also known as tailings [66]. After collection, sediment samples were stored in the dark at in situ temperature until further processing for the setup of bioremediation experiments. Additional aliquots of sediment samples were used for the analysis of heavy metal concentrations and their repartition in the different geochemical phases.

### 2.2. Sediment Remediation Experimental Setup, Microbial Strains Used, and pH Determination

Portman Bay sediments were subjected to different experimental treatments, including the addition of chemicals alone, as well as a combination of different microbial additions (Table 1). The experimental setup followed the procedures already described in one of our previous similar experiments [10] with proper modifications (namely, the additional treatments with fungi and a more complex array of chemical amendments).

The first set of sediment samples (Table 1) was added with different chemicals, including treatment with either glucose, elemental S, Fe, or a mix of Fe and elemental S, or of Fe and glucose. The second set of samples (Table 1) was added with bacteria only (Fe/S oxidizing chemo-autotrophic bacteria, chemo-heterotrophic bacteria, or a mix of both types of bacteria), also testing different types of contextual chemical amendments with S, Fe, and/or glucose. Finally, the third set of sediment samples (Table 1) was added with fungi (either alone or in association with Fe/S oxidizing bacteria), also supplementing glucose (potentially promoting fungal heterotrophic metabolism) and Fe (potentially promoting Fe/S oxidizing bacteria). A parallel set of sediments was incubated without amendments and used as controls.

As reported in Table 1, it can be noted that we used the same concentration of glucose (0.1 g/L) across the different treatments, which we previously tested to be optimal for the bacterial treatments [10], but can be considered somewhat lower compared to usual fungal culture media ([41,42] and ref. therein). Nevertheless, such relatively low glucose concentrations were selected based on several considerations. (i) to exclude possible inhibition of autotrophic bacteria and related biases due to higher glucose concentrations [67], (ii) to trigger organic matter priming in sediments [68], leading fungi to consume also other organic substrates available in sediments, so to enhance fungal-mediated dissolution of the metals possibly bound to the sedimentary organic fractions, (iii) glucose additions of 0.1 g/L have been already shown to trigger a relevant increase in fungal biomass in large-scale bioreactors [69]. (iv) as our fungal strains are of marine origin, we hypothesized that they could grow well also at glucose concentrations much lower than those used in laboratory cultures (i.e., in oligotrophic conditions typical of marine ecosystems). So, we conducted preliminary tests (see Appendix A), which show that all bacterial and fungal strains can grow well under the test conditions used in our study.

Acidophilic chemo-autotrophic bacteria (*A. thiooxidans* DSM 504, *A. ferrooxidans* DSM 14882T, and *L. ferrooxidans* DSM 2705T) and the acidophilic chemo-heterotrophic bacterial strain (*A. cryptum* DSM 2389T) were purchased in pure cultures at DSMZ and cultivated according to the standard supplier’s instructions (www.dsmz.de; [10]). Culture media consisted in DSMZ 35 medium for *A. thiooxidans* (0.10 g/L of NH4Cl, 3.00 g/L of KH_2_PO_4_, 0.10 g/L of MgCl_2_ 6H_2_O, 0.14 g/L of CaCl_2_ · 2H_2_O and 10 g/L of S_0_), DSMZ 882 medium for *A. ferrooxidans* and *L. ferrooxidans* (consisting of 950 mL of solution A: 0.139 g/L of (NH_4_)_2_SO_4_, 0.056 g/L of MgCl_2_ 6H_2_O, 0.028 g/L of KH_2_PO_4_, 0.155g/L of CaCl_2_ · 2H_2_O; 50 mL of solution B: 44.5 g/L of FeSO_4_ 7H_2_O; and 1 mL of solution C: 0.076 g/L of MnCl_2_ · 2H_2_O, 0.068 g/L of ZnCl_2_, 0.064 g/L of CoCl_2_ 6H_2_O, 0.031 g/L of H_3_BO_3_, 0.010 g/L of Na_2_MoO_4_, 0.067 g/L of CuCl_2_ · 2H_2_O), and DSMZ 269 medium for *A. cryptum* (2.0 g/L of (NH_4_)_2_SO_4_, 0.1 g/L of KCl, 0.655 g/L of K_2_HPO_4_·3H_2_O, 0.5 g/L of MgSO_4_·7H_2_O, 0.3 g/L of yeast extract, 1.0 g/L of glucose). For bacterial cultivations, all solutions were sterilized before use, and flasks were incubated at 35 °C on a rotary shaker at 150 rpm (Stuart orbital incubator S510).

Fungi belonging to *Aspergillus niger* and *Trichoderma sp*. were obtained following isolation in pure cultures from HM-contaminated marine sediments of the Bagnoli Bay (Tyrrhenian Sea, Naples, Italy) as previously described [70], by dilution plating technique with marine agar containing 0.3 mg/mL rifampicin to avoid bacterial growth and incubating at room temperature for 7 days [70]. These marine fungal strains were selected for our bioleaching experiments as belonging to fungal taxa previously described as able to decrease pH and enhance HM solubilization in bioleaching experiments in soils, freshwater sediments, and other matrices, and hence hypothesized to perform similarly for the decontamination of the marine sediments collected in the present study [16,71,72,73]. Indeed, *A. niger* has been previously exploited for the bioremediation of mine tailings [11], HM-contaminated freshwater sediments [71], and toxic industrial wastes [72]. Similarly, *Trichoderma* sp. has been shown to solubilize metals from soils and plant tissues [60,61,74] and was also proposed for HM bioremediation of marine environments (even if not directly tested; [73]).

All sediment remediation experiments were set up at 12.5% *w/v* (weight of the dry sediment to final volume) in autoclaved 250 mL Pyrex flasks, with 150 mL final volume. For the treatments with bacterial and/or fungal additions, bacteria and/or fungi were inoculated from cultures in exponential growth (15 mL of bacteria at a concentration of 1.5–2·10^8^ cells mL^−1^, and/or 10 mg of fungal biomass for each microcosm). All flasks were kept at a constant temperature of 20 °C on a rotary shaker (150 rpm) (Stuart orbital incubator S510), and all microcosms were set up in triplicate. At the beginning of the experiments, all microcosms were set at a sediment pH of 2.5, and pH values were checked during the experiments using an inoLab Multi 720 pH meter (WTW) equipped with a temperature probe (SenTix 81, WTW). Aliquots were collected from each experimental microcosm during the experimental incubations to determine HM concentrations, as described below.

### 2.3. Determination of HM Concentrations and HM Bioremediation Yields

Before starting the experiments, HM contents of the Portman Bay sediment, which have been used for the remediation experiments, were determined after acid digestion as previously described [70]. Briefly, sediment sub-samples were heated for 90 min at 150 °C in Teflon boxes, following addition with 5 mL fluoridric acid and 1 mL of HCl:HNO_3_ (3:1). Then, sediments were amended with 5 mL of 10% boric acid, and the extracts were assayed by atomic absorption spectrophotometry and by inductively coupled plasma-atomic emission spectrometry [70].

To assess the fractions of heavy metals associated with the different geochemical phases of the sediments, we adopted a selective extraction procedure, using specific chemical reagents to sequentially extract HMs in the following phases: (i) the exchangeable and carbonate bound fractions (hereafter, exchangeable fraction), extracted utilizing 0.11 M acetic acid, pH 2.8; (ii) Fe and Mn oxides fractions (reducible fraction), extracted with 0.1 M NH_2_OH, pH 2; (iii) organic and sulfide fraction (oxidizable fraction), extracted with hydrogen peroxide 30% and treated with ammonium acetate ((C_2_H_7_NO_2_) at pH 2, and (iv) the residual fraction, that remains in the residual solid, obtained by difference with total metal contents [70,75].

It is worth noting that sediments were homogenized before starting the experiments and before sampling for heavy metal analysis to obtain representative values for the total volumes of treated sediments.

On the basis of the chemical characterization, As, Cd, and Zn were identified as the main HMs in the Portman Bay sediments (see results and discussion section), and as such, we specifically assessed the leaching efficiency of these three elements by comparing their concentrations at the beginning and at the end of the incubation period following each different experimental treatment.

### 2.4. Statistical Analyses

To test for differences in the bioremediation yields obtained by the different experimental treatments, we used analysis of variance following homogeneity of variance checks by the Cochran’s test. Pair-wise tests were performed in case of significant (*p* < 0.05) differences. Statistical analyses were run using Primer 6 + PERMANOVA [76].

## 3. Results and Discussion

The analysis of HMs in the sediments of Portman Bay used for the bioremediation experiments revealed the presence of high HM concentrations, especially for Zn, As, and Cd (2963, 250, and 4.7 µg/g, respectively; Figure 1A).

Such concentrations exceed national and international sediment quality guidelines [66,77,78] and are above the ERL-“Effects Range Low” values and, for As and Zn, also above the ERM-“Effects Range Median” values, indicating possible or highly probable adverse ecotoxicological effects [79,80]. The sequential extraction procedure highlighted that Zn, As, and Cd were present in a different proportion associated with the different geochemical fractions of the sediments (Figure 1B). In particular, the largest fraction of the As was associated with the residual fraction (80%), whereas Zn was mostly associated with the exchangeable/carbonate fraction, followed by reducible and oxidizable fractions. Such a different repartition can influence to a great extent, the solubilization efficiency of the different elements since metals bound to the residual fraction is difficult to mobilize through bioleaching [1,81,82].

In this study, we found that sediments treated with glucose or S displayed solubilization efficiency of the three target elements very similar to those observed in untreated sediments (i.e., no-amendment; Figure 2A). Differently, the addition of Fe significantly increased the solubilization efficiency for all three elements, independently of the concomitant addition of S and/or glucose (Figure 2A), possibly due to Fe-induced lowering of sediment pH and consequent stimulation of the bioleaching ability of autochthonous microbial assemblages inhabiting the Portman Bay sediments [1,83].

The addition of Fe/S oxidizing bacteria alone had a limited solubilization efficiency, but their efficiency significantly increased in case of contextual addition of Fe, S, or both (up to 84.6 ± 2.8%, 66.9 ± 2.7%, and 8.9 ± 1.2% for Zn, Cd, and As, respectively; Figure 2A), likely due to their metabolic stimulation induced by such compounds [1,10,84,85]. We could notice that the addition of S stimulated Fe/S oxidizing bacteria less than the addition of Fe, suggesting a reduced ability of Fe/S bacteria in our experiments to enhance HM bioleaching by using S to generate sulphuric acid [86]. Our results also highlight the lack of synergistic effects of the concomitant addition of Fe and S on the bioleaching efficiency of Fe/S oxidizing bacteria, as previously documented [87]. Overall, the bioleaching yields obtained in this study confirm the generally good performance of Fe/S oxidizing bacteria in HM bioleaching from marine sediments, as reported in similar previous works [88,89].

The addition of the chemo-heterotrophic bacterium *A. cryptum* alone did not determine significant effects on the solubilization performance of HMs when compared with the controls (Figure 2A). Similarly, the combined addition of Fe/S oxidizing bacteria and *A. cryptum* resulted in similar or even lower bioleaching efficiency than those obtained with Fe/S oxidizing bacteria alone, further indicating that the *A. cryptum* was inefficient or even reduced HM solubilization in Portman Bay sediments. This result disagrees with that previously reported on sediment samples collected in another contaminated coastal site in which the combined addition of Fe/S oxidizing bacteria and *A. cryptum* determined an almost double bioleaching efficiency compared to that observed by using these two kinds of bacteria alone [10]. As also highlighted in other cases [1], HM solubilization in contaminated marine sediment and the outcome of bioremediation approaches largely depend on the specific characteristics of the sediments, which hampers to drawn definitive conclusions on their effectiveness.

The addition of fungi, either alone or in combination with bacteria, determined a significant increase in bioleaching efficiency compared to the controls, reaching values similar to (for Zn and Cd) or even higher than (for As) that were obtained when using Fe/S oxidizing bacteria (Figure 2A). The calculation of the microbial-induced increase in solubilization yield (i.e., by comparing each microbial-based treatment with the respective chemical-only treatment) allowed further highlighting the overall better performance of the fungal-based treatments compared to the addition of bacteria, especially for Zn and As (Figure 2B). In particular, the highest microbial-induced increase in solubilization yield was reached by the treatment with fungi alone (with significantly higher peaks for Zn and As) (Figure 2B). These results indicate that in the case of the Portman Bay sediments, the addition of selected fungal strains, especially without contextual bacterial addition, can represent the most promising strategy to enhance HM bioremediation efficiency. A similar study conducted on freshwater sediments [33] highlighted that the co-addition of fungi, able to consume organic compounds which would otherwise be toxic for autotrophic Fe/S oxidizing bacteria, could increase the bacterial bioleaching efficiency in HM-contaminated sediments. Our study thus extends the relevance of fungi beyond this proposed role of facilitators for bioremediating bacteria, as we show that the addition of fungi alone, without bacterial additions, can reach similar or even higher HM bioremediation efficiencies.

To our knowledge, only one study has investigated the use of marine-derived fungi for the removal of HMs from marine sediments so far [90]. However, this approach is very different compared with ours, as not based on fungal bioleaching processes. Rather, Cecchi et al.ii used a membrane enriched with marine fungi that bioaccumulate metals to be put in contact with the HM-contaminated sediments to be treated [90]. As such, the possibility of properly comparing results from the two approaches is limited. Indeed, applying Cecchi et al. system, only sediments that are proximal to the membrane can be treated, and the materials used need then to be collected and disposed of as special wastes after HM accumulation [90]. So, the two approaches can have different advantages or limits, and their application should be based on the final aim of each specific decontamination context.

Despite the lack of similar studies on marine sediments on the use of marine-derived fungi for the removal of HMs through bioleaching, our results can be compared with analogous experiments conducted on other matrices (e.g., river sediments, soils, solid wastes). For example, the overall bioleaching yields for Cd and Zn obtained by adding fungi on Portman Bay sediments are either higher or lower than what was previously reported by similar bioremediation experiments on freshwater sediments (i.e., only 2% [71] and up to >99% [64] for Cd, while 44% [71] and up to >80% for Zn [33]), and similarly for As (i.e., from 3% to 62% based on a study on mine tailings [91]). However, such comparisons should be viewed with caution. Indeed, differences in fungi-mediated bioleaching can depend on a wide array of factors, including the matrix type and geochemistry, the specific fungal strains utilized, the microbial interactions between autochthonous and inoculated microbes, and different susceptibility of the added microbes to the toxic contaminants present in the treated matrix [1,35,70,92]. Indeed, different studies have reported that different fungal strains have a different ability to thrive at high concentrations of heavy metals, which thus can result in a different bioleaching potential [34,58,93]. Our results based on multiple bioleaching experiments conducted in parallel provide further support that bioaugmentation approaches with fungi could be effective to the same extent or even more than those based on bacterial additions [64,94]. These results thus contribute to pointing out the rising role of fungi as efficient microbial taxa for HM bioremediation and also for marine sediments [41,42]. Notably, the normalization of the HM solubilization efficiency values to the HM extractable fraction (that is, to the overall HM mass, except the residual fraction; Figure 3) highlights that fungal addition was the only biotreatment able to overcome 100% solubilization yield for Zn and Cd (with values up to 106% and 126% for Zn and Cd, respectively). This implies that the treatment with fungi was the only one able to solubilize also a fraction of Zn and Cd bound to the residual fraction. Considering the relatively short time span of our experiments (14 days), these results led us to hypothesize a potentially even higher bioleaching yield over a longer period. Nevertheless, further studies are needed to assess the potential of the fungal strains tested in our experiments on the solubilization of the less mobile fraction of HMs bound to the residual fraction [1,95].

Preliminary tests we conducted to optimize the growth of the tested bacterial and fungal strains under our laboratory conditions indicated that the marine-derived fungal strains *A. niger* and *Trichoderma* sp. show higher biomass yields and hence possibly perform even better in terms of bioremediation efficiency using glucose concentrations higher than 0.1 g/L (Appendix A). Despite this, the use of low glucose concentrations can be a valuable compromise to promote HM solubilization by minimizing possible glucose-induced inhibition of autotrophic bacteria in co-culture with fungi [96], as well as by reducing the overall carbon footprint [97,98]. We thus suggest that future tests should be carried out to assess the minimum glucose concentration to effectively sustain microbial growth rates and HM bioleaching efficiency in order to improve the overall eco-sustainability of the bioremediation process.

The analysis of sediment pH in each of the experimental microcosms at the end of the incubation period (Figure 4) highlighted significant differences among treatments and among treatments and the initial pH value of 2.5 of the sediment (which had been standardized across all microcosms at the beginning of the experiments, see methods section). In particular, the controls (without any amendment) were characterized by the highest pH values, which were similar to those observed in the case of the addition of glucose, sulfur, Fe/S oxidizing bacteria, or glucose + *A. cryptum* (overall pH values ranging from 4.0 ± 0.2 to 4.9 ± 0.4). The chemical-based treatments, as well as the treatment based on the addition of Fe/S bacteria without Fe or with *A. cryptum*, displayed at the end of the experiments pH values similar to those measured at the beginning (i.e., ranging from 2.5 to 3.1). In these microcosms, we can hypothesize that the addition of Fe contributes to keeping lower pH values over time, possibly via the thiosulfate and/or polysulfide pathways or similar Fe-dependent mechanisms able to generate protons [1,83]. Conversely, the microcosms with the addition of Fe/S bacteria and Fe, as well as those with the addition of fungi, were characterized by the lowest sediment pH values (ranging from 1.7 to 2.2), significantly lower even when compared to the pH determined at the beginning of the experiments (2.5).

Notably, we found significant and positive relationships (Figure 4) between the solubilization efficiency of all HM analyzed and sediment pH (that is, the lower the sediment pH reached by the experimental treatment, the higher the Zn, Cd, and As solubilization yields obtained). From these relationships, we can observe that, at similar low pH values (<2.5), the addition of fungi provides a better bioleaching performance for As compared to the addition of Fe/S oxidizing bacteria (Figure 4), suggesting that specific metabolic features of *A. niger* and *Trichoderma* sp. fungal strains could lower sediment pH and enhance As bioleaching. Indeed, even in the absence of Fe addition, fungi were able to keep the lowest pH values over time, hence suggesting a Fe-independent mechanism able to keep acid pH [1,83]. Our results agree with previous studies reporting a marked increase in HM solubilization efficiency under more acidic conditions [99,100,101]. Indeed, low pH values not only directly enhance chemical solubilization but also boost metabolic and overall growth rates of acidophilic microbes [99,100,101]. Moreover, acidophilic microbes, including the Fe/S oxidizing bacterial and fungal strains tested in our experiments, are known to contribute to sediment acidification by producing H_2_SO_4_ and Fe^3+^ [46,102] and/or organic acids such as tartaric, oxalic, citric, and malic acids [64,102,103,104,105,106]. Even if not specifically investigated in this study, such microbial processes and secondary metabolites may be involved in explaining the high bioremediation yields we observed. However, further studies are needed to unveil which specific metabolic features could contribute to sediment acidification in our treatments (e.g., which kind of organic acids can be produced by the tested fungal strains). In addition, as we only assayed the use of a mix of the two different fungal strains in our tests, further tests with *A. niger* or *Trichoderma* sp. alone could disentangle their relative role in the bioremediation potential observed in our study.

## 4. Conclusions

In our study, we compared different approaches for the remediation of marine sediments highly contaminated by HMs, based on chemical treatments and/or different microbial-based amendments with bacteria and/or fungi. We show that fungal additions can result in HM bioleaching yields similar, or significantly higher, than those obtained by chemical or bacterial treatments. Moreover, we show that fungi could outperform bacteria in the bioleaching of the less mobile HM fraction. Overall, our findings indicate that fungi can be more effective than acidophilic autotrophic and heterotrophic bacteria in HM bioleaching, and as such, their use can represent an alternative strategy for the bioremediation of marine sediments highly contaminated with heavy metals.

## Figures and Tables

**Figure 1 microorganisms-10-00993-f001:**
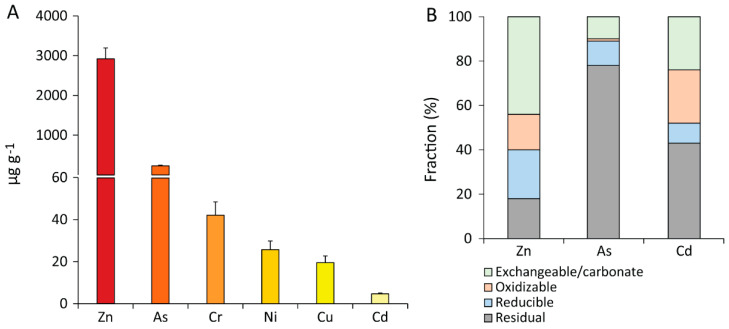
Total concentrations of metals and their repartition into the different geochemical phases in the sediments of Portman Bay. In the left panel (**A**), reported are the concentrations of the main HMs found in the sediments collected for the setup of the bioremediation experiments. In the right panel (**B**), reported is the partitioning of each HM in the four geochemical fractions (exchangeable, oxidizable, reducible, residual) for the three elements found to exceed sediment quality guidelines and hence further selected as targets for our bioremediation tests. Reported are average values and SDs.

**Figure 2 microorganisms-10-00993-f002:**
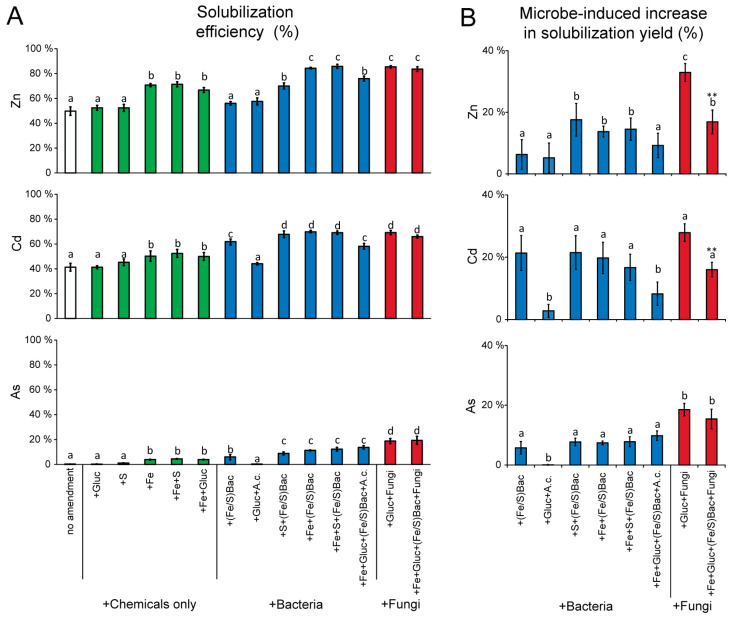
Bioremediation performance of the different combinations of chemical and microbial amendments and controls performed in this study. The left panel (**A**) shows the solubilization efficiency for Zn, Cd, and As. The right panel (**B**) shows the increase in solubilization yields induced by the addition of each different type of microbes (calculated by comparison with the respective chemical-only treatments without microbial addition). +Gluc: glucose addition; +Fe: iron addition; +S: sulphur addition; + Fe/S Bac: addition of chemo-autotrophic Fe/S oxidizing bacteria (including *Acidithiobacillus ferrooxidans*, *Acidithiobacillus thiooxidans,* and *Leptospirillum ferrooxidans*); +A.c.: addition of the chemo-heterotrophic bacteria (*Acidiphilium criptum*); + Fungi: addition of fungal strains (*Aspergillus niger* and *Trichoderma* sp.). Different letters in the bar charts highlight significant differences among values. In the (**B**) panel, asterisks highlight significant differences between the values obtained with the addition of fungi alone, or of fungi and Fe/S bacteria.

**Figure 3 microorganisms-10-00993-f003:**
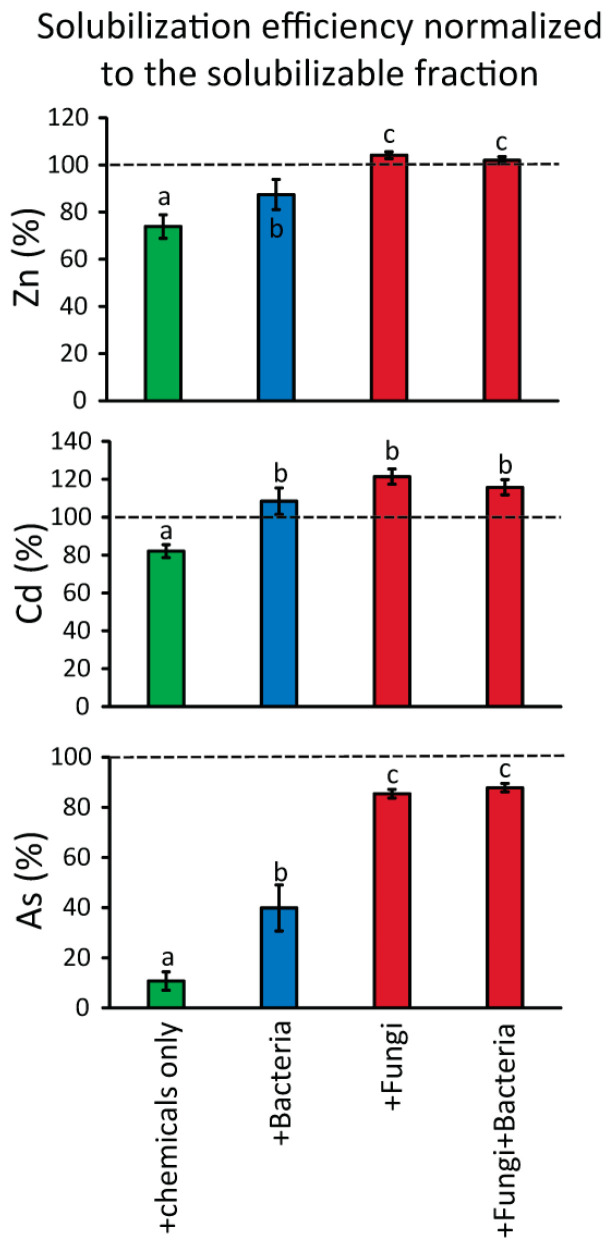
Solubilization efficiency normalized to the extractable fraction of Zn, Cd, and As. Reported are the solubilization efficiency values obtained by the four main different conditions used in this study: chemicals only, bacteria, fungi, and the mix of bacteria and fungi. Different letters in the bar charts highlight significant differences among values.

**Figure 4 microorganisms-10-00993-f004:**
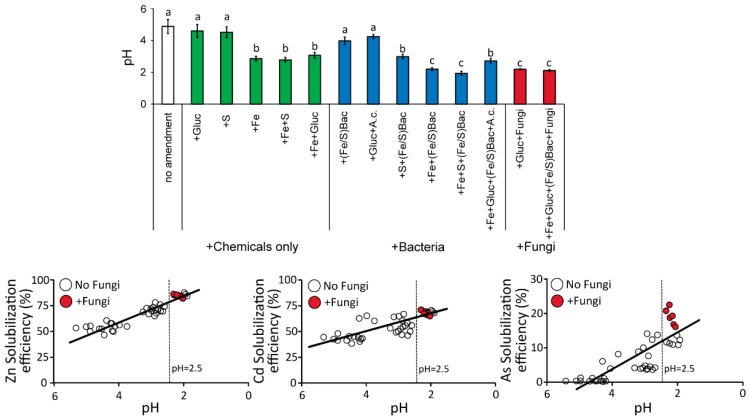
Sediment pH values among experimental treatments and controls at the end of the incubation period and their correlation with HM solubilization efficiency. In the upper panel, reported are average sediment pH values and SDs. In the lower panel, reported are the relationships between the Zn, Cd, or As solubilization efficiency and sediment pH, with the white dots indicating overall treatments without fungal additions and red dots representing the overall treatments with the addition of fungi. Different letters in the upper bar chart highlight significant differences among values.

**Table 1 microorganisms-10-00993-t001:** Description of the different combinations of chemical and microbial amendments performed in this study. Reported are also the final concentrations for the chemical amendments. +Gluc = glucose addition; +Fe: iron addition; +S: sulphur addition; + Fe/S Bac: addition of chemo-autotrophic Fe/S oxidizing bacteria (including *Acidithiobacillus ferrooxidans*, *Acidithiobacillus thiooxidans*, and *Leptospirillum ferrooxidans*); +A.c.: addition of the chemo-heterotrophic bacteria (*Acidiphilium criptum*); + Fungi: addition of fungal strains (*Aspergillus niger* and *Trichoderma* sp.).

Treatment	Sample ID	Fe (g/L)	S (g/L)	Glucose (g/L)
Incubation of original sediments (no additions)	No amendement	0	0	0
Addition of Chemicals Only	+Gluc	0	0	0.1
	+S	0	10	0
	+Fe	4.5	0	0
	+Fe+S	4.5	10	0
	+Fe+Gluc	4.5	0	0.1
Addition of Bacteria (autotrophic Fe/S oxidising bacteria -(Fe/S) Bac-, and/or heterotrophic bacteria -A.c., *Acidiphilium cryptum*-)	+(Fe/S) Bac	0	0	0
	+Gluc+A.c.	0	0	0.1
	+S+(Fe/S) Bac	0	10	0
	+Fe+(Fe/S) Bac	4.5	0	0
	+Fe+S+(Fe/S) Bac	4.5	10	0
	+Fe+Glu+(Fe/S) Bac+A.c.	4.5	0	0.1
Addition of Fungi	Gluc+Fungi	0	0	0.1
Addition of Fungi and Fe/S oxidising bacteria	+Fe+Glu+(Fe/S) Bac+Fungi	4.5	0	0.1

## Data Availability

All data needed to evaluate the conclusions of this work are present in the paper and the Appendix A.

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
