# Peer review of "Fungi Can Be More Effective than Bacteria for the Bioremediation of Marine Sediments Highly Contaminated with Heavy Metals"

_microorganisms, 2022, doi:10.3390/microorganisms10050993_

Round 1
Reviewer 1 Report
Please answer the following:
- have you analysed the evolution in the number of microorganisms? Bioremediation sometimes generates long lag phases, fungi grow slower than bacteria. During the 14 days for the experiments were the microorganism analysed ?
- have you analysed the glucose variation during the 14 days of the experiments? Very low concentration of glucose are provided: 0.1 g/l for a mixture of bacteria and fungi. Have you analysed if the glucose was consumed? or the moment when it was consumed.
- toxic metals may show different gradient distributions over very small interfaces. Please provide more information on the sampling technology (referring to heterogeneity of the sediments in vertical and horizontal dimensions).
- some reactions between sediments-metals-microbial interactions would be of interest
- explain the choice of the fungal strains. why not chose marine fungi?
- have you analysed the acids that lower the pH ?
- please add a comparison between you results and other recent studies (https://doi.org/10.1016/B978-0-323-85455-9.00028-X)
- the references require more current references (within the last 5 years)
Author Response
REV #1
REV#1: Please answer the following: have you analysed the evolution in the number of microorganisms? Bioremediation sometimes generates long lag phases, fungi grow slower than bacteria. During the 14 days for the experiments were the microorganism analysed?
REPLY: First of all, we would like to thank Rev#1 for her/is useful comments and hints for deeper discussion of our results. Despite we found some of the aspects raised by Rev#1 to go somehow beyond the scope of our manuscript, we anyway exploited all the comments received. Accordingly, in the amended version of our manuscript, we provided additional data based on our available information, as well as acknowledgements of the further aspects and related analyses that could be conducted in similar future experiments to make them more informative.
This said, we thank REV #1 for this first question about the evolution of microbial abundance during our experiment. As explained in our original manuscript, the initial abundance/biomass was standardized across all treatments we performed. Moreover, we checked bacterial/fungal abundances over time during preliminary tests, even though a complete monitoring of microbial abundances during the incubations was not the focus of our manuscript (so, we initially decided not to include this partial dataset). To satisfy this comment by Rev#1, in the amended version of our manuscript we have provided our available additional data (and the related Supplementary Figure 1), which demonstrate growth of bacteria and fungi over time under our test conditions. We also included in this figure available data that deal with the following comment by Rev#1 regarding glucose concentrations, as explained in our next reply.
Moreover, in the amended version of our manuscript, we have implemented the discussion to underline the importance of monitoring microbial abundances over time in similar experiments.
REV#1: have you analysed the glucose variation during the 14 days of the experiments? Very low concentration of glucose are provided: 0.1 g/l for a mixture of bacteria and fungi. Have you analysed if the glucose was consumed? or the moment when it was consumed.
REPLY: We thank REV #1 for this question, which allowed us to deepen the methods section and the discussion section, by better explaining our rationale on this aspect. We selected glucose concentrations based on several considerations, which we detail here below, and we have better explained in the amended version of our manuscript to avoid misunderstanding.
- Higher glucose concentrations may inhibit autotrophic bacteria (see for example Marchand and Silverstein, 2003), thus introducing possible biases.
- Fungi are known to respond to glucose additions at such concentrations of 0.1 g/L (Wang et al 2019).
- Fungi can consume not only the added glucose, but also the organic matter present in the sediments. So, the glucose added was not the only possible carbon source for fungi in our experimental system. Glucose addition in our experiment was intended to stimulate fungal growth as well as possible priming effects (Guenet et al., 2010) promoting fungal utilization of the sedimentary organic matter, to enhance fungal-mediated dissolution of the metals possibly bound to the organic fractions. At the same time, a detailed investigation of such processes of organic C consumption during our experiment was beyond the scope of our work, so we did not include specific analyses for this aspect.
- Concentrations of 0.1 g/L are optimal for A cryptum based on our previous experience with tests under different glucose concentrations (Beolchini et al., 2009), and we chose to use the same glucose concentrations across the different treatments. This, to avoid possible confounding effects due to different glucose concentrations, and at the same time keeping the number of experimental systems at a level that was possible to manage in our laboratory.
- We have evidence that our marine fungal isolates are able to grow well at 0.1 g/L glucose concentrations in the experimental mesocosms. We can speculate that our fungal strains, isolated from marine sediments, may be adapted to live under more oligotrophic conditions than other fungal strains grown in high-glucose laboratory culture broths. In the amended version of our manuscript, we have included additional available data (Supplementary Figure 1) on fungal abundance over time using 1 g/L or 0.1 g/L of glucose, based on our preliminary tests. Even if partial, these data provide useful information related to this comment by Rev#1, showing that the bacterial and fungal strains can grow well under the test conditions and allowing us to hypothesize that even higher fungal proliferation and bioremediation yields may be obtained under glucose concentrations higher than those tested in our study. Accordingly, we have deepened the discussion on this point in our amended version of the manuscript.
- In our research, we usually pay particular attention to the eco-sustainability of the proposed bioremediation approaches. Indeed, glucose is a relatively bad carbon source in terms of non-renewable energy use and greenhouse gas emissions based on LCA (Life Cycle Assessment) studies, so low-glucose consuming processes and/or alternative C sources can be considered as more eco-sustainable, as able to minimize the overall C footprint (Tsiropoulos et al 2013; Dhillon et al., 2011). As such, the selection of bacterial/fungal strains that can grow well under oligotrophic conditions, as in our case, can be considered a positive feature for bioremediation. We have added discussion about this in the amended version of our manuscript.
Finally, we acknowledge that in the present work, we did not determine glucose concentrations over time during the experiments. We agree that, even if beyond the scope of our work, this information may help in optimizing the biotreatment performance in the future, so in the amended version of the manuscript we have added a sentence also about this.
References cited in our reply:
Marchand, E. A., & Silverstein, J. (2003). The role of enhanced heterotrophic bacterial growth on iron oxidation by Acidithiobacillus ferrooxidans. Geomicrobiology Journal, 20(3), 231-244.
Wang, S., Liu, P., Shu, W., Li, C., Li, H., Liu, S., ... & Noorman, H. (2019). Dynamic response of Aspergillus niger to single pulses of glucose with high and low concentrations. Bioresources and Bioprocessing, 6(1), 1-14.
Guenet, B., Danger, M., Abbadie, L., & Lacroix, G. (2010). Priming effect: bridging the gap between terrestrial and aquatic ecology. Ecology, 91(10), 2850-2861.
Tsiropoulos, I., Cok, B., & Patel, M. K. (2013). Energy and greenhouse gas assessment of European glucose production from corn–a multiple allocation approach for a key ingredient of the bio-based economy. Journal of cleaner production, 43, 182-190.
Dhillon, G. S., Brar, S. K., Verma, M., & Tyagi, R. D. (2011). Utilization of different agro-industrial wastes for sustainable bioproduction of citric acid by Aspergillus niger. Biochemical Engineering Journal, 54(2), 83-92.
REV#1: toxic metals may show different gradient distributions over very small interfaces. Please provide more information on the sampling technology (referring to heterogeneity of the sediments in vertical and horizontal dimensions).
REPLY: We thank REV#1 for this comment, that allows us to better explain this point. Actually, our experimental systems were relatively small and sediments were homogenized before starting the experiments and sampling for heavy metals, so to obtain representative values for the total volumes of treated sediments. In the amended version of the manuscript, we have better detailed this aspect.
REV#1: some reactions between sediments-metals-microbial interactions would be of interest. Explain the choice of the fungal strains. why not chose marine fungi?
REPLY: We thank Rev#1 for these comments. We agree that a more specific investigation of the sediment-metals-microbial interactions could be interesting in the framework of our experiments, but we are sorry if in this case we could not assess these aspects in higher detail. Regarding the fungal strains used in our experiments, these were actually fungi of marine origin, isolated from marine sediments, as already stated in our original manuscript version. To avoid misunderstanding, we modified several sentences throughout the text, so that now their marine origin is more clear.
REV#1: have you analysed the acids that lower the pH ?
REPLY: We thank Rev#1 for pointing out this aspect, which unfortunately could not be considered in our experiment. Similarly to other previous comments by Rev#1, these aspects, despite interesting and worth investigating, go beyond the scope of the present manuscript. Surely, we exploited them to provide insights on the further aspects and related analyses that could be conducted in similar future experiments to make them more informative, by adding discussion and related references in the amended version of our manuscript.
REV#1: please add a comparison between you results and other recent studies (https://doi.org/10.1016/B978-0-323-85455-9.00028-X).
REPLY: We thank Rev#1 for this request. We have added also this article in the reference list and exploited it for implementing the discussion section. We would like to note that we added several other recent papers and we enriched and updated the introduction and discussion to a very larger extent.
REV#1: the references require more current references (within the last 5 years)
REPLY: We thank Rev#1 for this comment and in the amended version of the manuscript we included . several other recent papers and we enriched and updated the introduction and discussion to a very larger extent.

Reviewer 2 Report
The revised manuscript studies bioremediation of marine sediments containing some HMs. This topic is always important.
Generally, work is quite well described, well organized and presented.
It could be, however, improved. The main topic of the study are fungi, but only two treatments involve them. In the introduction and discussion the fungi are less described. These parts could be improved.
I wonder what was the rationale to study role of fungi? They were isolated from other sediments (two strains), the same for bacteria. Was there microbial diversity of the studied sediments determined? There should be more explanation for the choice of microbes and the need for studying marine sediments.
Authors often refer to their previous work what makes a little problematic to follow this manuscript. For me experimental design is not full described. I miss media for growing bacteria/fungi, more detailed experiment description (e.g. in which vessels the experiment was conducted, time etc.).
One mistake in line L135 - upper index is missing in "2 · 108 cells ml"
Figures are very good, only the font is too small and one needs to use 150% magnification to read them well.
I also wonder about the efficiency of studied microorganisms - only solubililzation was tested - who does it determine bioremediation efficiency - did you tested metal removal or accumulation in microbial biomass?
How would you prove the advantage fungi over bacteria - on Fig. 2 it is the same for both groups? Could you comment more on this issue?
Author Response
REV #2
REV#2: The revised manuscript studies bioremediation of marine sediments containing some HMs. This topic is always important. Generally, work is quite well described, well organized and presented. It could be, however, improved.
REPLY: We thank Rev#2 for the very positive comments and hints to improve our manuscript. We have carefully followed all her/his comments, as detailed below.
REV#2: The main topic of the study are fungi, but only two treatments involve them. In the introduction and discussion the fungi are less described. These parts could be improved.
REPLY: We thank Rev# for this comment. Accordingly, we have largely improved the introduction section with additional information and related recent references on fungi. Regarding the number of treatments with fungi, we agree that additional types of treatment could allow obtaining additional information (for example treatments with the two fungal strains used in parallel and not in mix). Even if we could not perform additional treatments with fungi in this work, we have added a sentence in the discussion section of our amended manuscript with some insights on interesting additional treatments that could be tested in the future.
REV#2: I wonder what was the rationale to study role of fungi? They were isolated from other sediments (two strains), the same for bacteria. Was there microbial diversity of the studied sediments determined? There should be more explanation for the choice of microbes and the need for studying marine sediments.
REPLY: We thank Rev#2 for this comment. Accordingly, in the amended version of the manuscript we have largely implemented the introduction and methods section with more explanation for the choice of microbes and the need for studying marine sediments. Unfortunately, we could not analyse the microbial diversity in the sediments used for our experiment, anyway we feel that such information, even if interesting, can be considered somehow beyond the scope of our lab tests in this specific case.
REV#2: Authors often refer to their previous work what makes a little problematic to follow this manuscript. For me experimental design is not full described. I miss media for growing bacteria/fungi, more detailed experiment description (e.g. in which vessels the experiment was conducted, time etc.).
REPLY: We can understand this comment by Rev#2. Several information related to methodological aspects was already available in previous papers, so we omitted it in our first version to avoid repetition/duplication with existing literature (which is checked systematically during the Microorganisms editorial process). Nevertheless, to accomplish this request by Rev#2, we provided additional methods information in the amended version of our manuscript about the media used and more detailed experiment description (Line 144-162; Line 130-139). At the same time, we carefully reduced repetition/duplication in the methods description, as contextually requested by the Journal Editors within this same review round.
REV#2: One mistake in line L135 - upper index is missing in "2 · 108 cells ml"
REPLY: Corrected, thanks.
REV#2: Figures are very good, only the font is too small and one needs to use 150% magnification to read them well.
REPLY: Thanks for this very positive feedback on our Figures. We have increased the fonts as requested, thanks for this notice.
REV#2: I also wonder about the efficiency of studied microorganisms - only solubililzation was tested - who does it determine bioremediation efficiency - did you tested metal removal or accumulation in microbial biomass?
REPLY: We thank Rev#2 for this comment. In the amended version of our manuscript, we have better clarified that we tested metal removal through bioleaching processes in our experiments (not metal accumulation in microbial biomass). This also allowed us to deepen the discussion and comparison of our work and results with other relevant studies.
REV#2: How would you prove the advantage fungi over bacteria - on Fig. 2 it is the same for both groups? Could you comment more on this issue?
REPLY: We thank Rev#2 for this comment. We feel that confusion on this point was mainly generated by our not optimal choice on the editing of Figure 2, which had too many panels and could result a bit complex. Thus, we better separated Figure 2 into Figure 2a and Figure 2b, so that now it is more clear to which panel the sentences of the results and discussion section refer to. Indeed, we can state that, overall, the highest values were obtained by the fungi-only treatment (especially considering the values in Figure 2b, after calculation of the microbial-induced increase in solubilization yield and/or the values in Figure 3 after the normalization of the HM solubilization efficiency values to the HM extractable fraction). At the same time, we agree with Rev#2 that, for Cd (but only for Cd), the highest peaks obtained with the fungal treatments are overall not statistically different from the values obtained with bacteria only (as it was already evident in our original Figures 2 and 3 based on the PERMANOVA analysis we showed). We would like to note that, despite a less robust statistical approach (T test) allows appreciating a significant difference also for Cd between fungi-only and bacteria-only treatment in Figure 3, we decided to keep the PERMANOVA results to be more robust. In the amended version of our manuscript, we modified this part acknowledging that for Cd the obtained higher peaks are actually not statistically different between the two treatments (fungi vs bacteria-only additions) (Line 289).
We want to highlight that this slight change in the results & discussion section does not modify our original main conclusion, expressed also in the title of our manuscript, that fungi “can be” more effective than bacteria (that is, not in all cases and for all metals).

Reviewer 3 Report
Manuscript entitled “Fungi can be more effective than bacteria for the bioremediation of marine sediments highly contaminated with heavy metals” submitted by Filippo Dell’Anno, Eugenio Rastelli, Emanuela Buschi, Giulio Barone, Francesca Beolchini and Antonio Dell’Anno, can be considered for publication in Microorganisms Journal, after a serious major revision.
Here is a list of my specific comments:
- General comment: The utility of this study should be clearly highlighted in the manuscript.
- Title: Replace title by “Comparative study of fungi and bacteria efficiency for the bioremediation of marine sediments highly contaminated with heavy metals” (or similar).
- Page 1, Abstract: Include in this section the most important experimental results to highlight the importance of this study.
- Page 1, 1. Introduction: This section is too brief and should be detailed. Pay attention on the most important aspects related to this topic and provide a clear description of the state of art in this field. May be the following paper: doi.org/10.1016/j.jenvman.2018.07.066 can be helpful for this.
- Page 3, 2.2. Sediment remediation experimental setup: This section should be reorganized. Pay attention on technical details and provide a clear description of the experimental methodology used in this study. The quantitative parameters evaluated in this study should be also mentioned.
- Page 4, 3. Results and discussion: In this section, all experimental results should be clearly presented and detailed discussed in accordance with the main objectives of this study.
- Page 9, Conclusions: This section is missing.
Author Response
REV #3
REV#3: Manuscript entitled “Fungi can be more effective than bacteria for the bioremediation of marine sediments highly contaminated with heavy metals” submitted by Filippo Dell’Anno, Eugenio Rastelli, Emanuela Buschi, Giulio Barone, Francesca Beolchini and Antonio Dell’Anno, can be considered for publication in Microorganisms Journal, after a serious major revision.
REPLY: We thank Rev#3 for her/his comments on our manuscript, which we have carefully considered in order to improve our manuscript for publication. Please find here below our point-by-point reply.
REV#3: Here is a list of my specific comments:
- REV#3: General comment: The utility of this study should be clearly highlighted in the manuscript.
REPLY: We thank Rev#3 for this comment. We have implemented the introduction section of our amended manuscript to make the importance and utility of our study clearer to the general reader.
- REV#3: Title: Replace title by “Comparative study of fungi and bacteria efficiency for the bioremediation of marine sediments highly contaminated with heavy metals” (or similar).
REPLY: In this case we would like to express our doubts about the opportunity of this change in the title proposed by Rev#3, which we feel would sound a bit too general and do not inform the reader about the originality and main result obtained in our experiments. Moreover, this proposed change seems to contradict her/his following point n.3 (i.e., Rev#3 request to make our main results more clear). For these reasons, we would like to keep the title proposed in our original version. Obviously, we remain available in case of different indication by Microorganisms Editorial Staff.
- REV#3: Page 1, Abstract: Include in this section the most important experimental results to highlight the importance of this study.
REPLY: We thank Rev#3 for this comment. Accordingly, we have modified some sentences of the abstract to make the importance of our results clearer to the general reader.
- REV#3: Page 1, 1. Introduction: This section is too brief and should be detailed. Pay attention on the most important aspects related to this topic and provide a clear description of the state of art in this field. May be the following paper: doi.org/10.1016/j.jenvman.2018.07.066 can be helpful for this.
REPLY: We agree with Rev#3 that the introduction section could be improved. Accordingly, we have exploited the proposed reference by Rev#3 as well as additional relevant and recent references, also considering suggestion by the other reviewers about the introduction section.
- REV#3: Page 3, 2.2. Sediment remediation experimental setup: This section should be reorganized. Pay attention on technical details and provide a clear description of the experimental methodology used in this study. The quantitative parameters evaluated in this study should be also mentioned.
REPLY: Also based on some other comments received by the other reviewers, we agree with Rev#3 that the methods section could be improved. Accordingly, we have reorganized and implemented this section with additional details on the experimental setup and the quantitative parameters determined in our study.
- REV#3: Page 4, 3. Results and discussion: In this section, all experimental results should be clearly presented and detailed discussed in accordance with the main objectives of this study.
REPLY: We thank Rev#3 for this general comment. We believe that, also based on the more specific comments received by the other reviewers, all the sections of our amended version of the manuscript are now significantly improved
- REV#3: Page 9, Conclusions: This section is missing.
REPLY: Based on our knowledge, Microorganisms accepts free formats, and several published papers in this Journal do not include a conclusions section. However, to accomplish Rev#3 request, in the amended version of the manuscript we have added this part. We agree that this section allowed to better highlight the main conclusions obtained in our work and we thank Rev#3 for this useful hint.

Round 2
Reviewer 1 Report
The article can be accepted in present form.
Reviewer 2 Report
Dear Authors,
as you adapted your manuscript according to my comments I can approve it now and recommend for publication
Best regards
Reviewer 3 Report
All my previous remarks and comments have been considered in this new version of the manuscript. In my opinion, the revised manuscript meets the criteria and can be published as review paper in Microorganisms Journal.